# Does blockchain technology matter for supply chain resilience in dynamic environments? The role of supply chain integration

**Abdullah Kaid Al-Swidi**[1], **Mohammed A. Al-Hakimi**[2]*, **Hussam Al Halbusi**[3], **Jaithen Abdullah Al Harbi**[4], **Hamood Mohammed Al-Hattami**[5,6]

1 College of Business and Economics, Qatar University, Doha, Qatar, 2 Marketing and Production Department, Thamar University, Dhamar, Yemen, 3 Management Department, Ahmed Bin Mohammed Military College, Doha, Qatar, 4 Imam Mohamed bin Saud Islamic University, Riyad, Saudi Arabia, 5 College of Business Administration, A'Sharqiyah University (ASU), Ibra, Oman, 6 Department of Accounting, Faculty of Commerce and Economic, Hodeidah University, Al Hudaydah, Yemen

* alhakimi111@gmail.com

**Data Availability Statement:** All relevant data are within the paper and its Supporting information files.

## Abstract

This study aims to empirically investigate the effect of blockchain technology (BCT) adoption on supply chain resilience (SCR), with the mediating role of supply chain integration (SCI) and the crucial effect of environmental dynamism (ED) as a moderator. Based on data collected from firms operating in the automotive industry in India, the proposed model was tested using Partial Least Squares Structural Equations Modelling (PLS-SEM) via SmartPLS software. The empirical results showed a positive effect of BCT on SCI, which in turn affects SCR. Importantly, SCI acts as a full mediator in the BCT-SCR relationship, which is moderated by ED, that is, the effect of BCT on SCR via SCI is strong when ED is high. This study offers the groundwork for operationalizing BCT in a supply chain context. It also contributes to SCR research by investigating how SCI mediates the effect of BCT on SCR. In addition, this study found a moderating effect of ED on the relationship between BCT and SCI. These results provide insights to auto manufacturers on ways to enhance SCR and ensure safe supply chain operations.

## 1 Introduction

In the present interconnected global market, uncertainties and disturbances pose unpredictable challenges to long-term success and sustainability [1], which overthrow traditional management practices that focus only on stable conditions [2]. Each day, companies face disturbances that can undermine their operational efficiency. One example of those threats is the COVID-19 pandemic, which has recently negatively impacted the global and inter-twined supply networks [3, 4]. Due to the disturbances in the supply chains (SCs), the Indian automotive industry has suffered severe production disruptions in many factories [5]. The primary factor behind this impact was the heavy reliance of India on China for obtaining auto components [6]. The outbreak of the coronavirus not only affected the automobile industry but also had a significant impact on the automotive components and forging industries. China holds a

**Funding:** The authors received no specific funding for this work.

**Competing interests:** The authors have declared that no competing interests exist.

dominant position as the leading supplier of auto components in India, with 27% of its exports. With the manufacturing facilities in China being temporarily shut down during the coronavirus crisis, numerous Indian automobile companies experienced substantial losses. Major companies like Tata Motors, Mahindra and Mahindra, and MG Motors in India have publicly acknowledged facing challenges in sourcing auto components from China, which has been severely affected by the virus [7]. Accordingly, Indian firms had to reconsider the structure of their SCs and how they should proceed in the future to predict, sense, and respond to future unexpected risks and crises in order to mitigate their impact.

Under these circumstances, companies need to build resilience that aids in "being alert to adapt to and respond to changes brought by a supply chain disruption effectively and efficiently [8]." The World Economic Forum [9] indicated that "more than 80% of firms place a strong emphasis on resilience to disruptions." Supply chain resilience (SCR)- which refers to, "the adaptive capability of the SC to prepare for unexpected events, respond to disruption and recover from them by maintaining continuity of operations at the desired level of connectedness and control over structure and function" [10]- is an essential dynamic capability (DC) in facing disturbances [11]. Nonetheless, the massive volume of disruptions that firms encounter may render it impractical to rely solely on internal resources or capabilities in the long run.

The latest advancements in the area of information and communication technologies (ICT) have emphasized the importance of SC digital twins and digital information technologies in managing SC disruption risks [12, 13] and making the SC more resilient [14–16]. Moreover, previous research has demonstrated that ICT enhances supply chain integration (SCI) by effectively managing the increased volume and intricacies of information exchanged among various SC partners [e.g., 17]. In addition, there are expectations that Industry 4.0 technologies enabled by ICT will further enhance process integration, thereby strengthening SCR [14]. Among these technologies, blockchain stands out as a notable solution with significant potential for addressing the complexities of SCs [18–21]. Blockchain Technology (BCT) is "an organizational capability that integrates all the SC assets and resources, adding value to the activities such as product tracking, information sharing, and providing transparency in SC transactions" [17]. BCT reflects a firm's ability to incorporate an ICT background in manufacturing [22] that helps firms to achieve the efficient coordination and synchronization of efforts necessary to develop SCR. With the exception of anecdotal evidence, however, the previous literature has been muted on the role of BCT, which enables firms to share information in an entirely secure and transparent manner, hence improving SCR. It is anticipated that BCT will have a substantial effect on SC processes within the automotive industry context [17]. Furthermore, as far as we know, there is a lack of empirical evidence regarding the effect of BCT on SCR in the automobile industry, necessitating further investigation of the role of BCT adoption in SC.

Even though some prior studies have provided a rationale for investing in ICT to improve SCI [e.g., 23] and SCR [24, 25], other research has demonstrated that these investments have not effectively generated an effect on organizational resilience [26]. These discrepancies in the findings serve as an impetus for us to further explore the connection between digital technologies, which support SCI, and SCR. According to various studies [e.g., 17], SCI is an important factor in enhancing collaboration and partnerships within the SC. The adoption of BCT enables secure storage of all supply chain transactions, and easy access to all partners, enhancing the level of SCI [27]. Previous research [e.g., 28] has also recognized SCI as an important variable that mediates the association between independent and dependent variables in the field of operations management. BCT has the potential to augment SCI and thereby contribute to SCR. Therefore, the current study aims to explore whether SCI acts as a mediator in the BCT-SCR relationship.

By embracing the dynamic capabilities theory (DCT), this study visualizes BCT as a dynamic capability and examines its direct effect on SCR, as well as its indirect effect through SCI. However, evidence from other research reveals that firms interested in adopting BCT must take into account the external context effect on the motivation to use technology [29]. Thus, environmental dynamism (ED) is a critical situational parameter in DCT, implying that the variation of competitive edge gained by organizational capability exploitation is contingent on ED [30]. This perspective is exemplified by contingency theory (CT). However, Eckstein *et al.* [31] contend that conceptual and empirical research on SC capabilities has mainly disregarded the influence of pertinent contextual factors. Furthermore, Clohessy and Acton [32] assert that empirical research on BCT has mainly overlooked the effect of ED. Previous research demonstrates unequivocally that a tumultuous outer environment may either boost or degrade a firm's most vital capabilities [e.g., 33]. As a result, evaluating the effect of BCT under different levels of ED remains challenging, indicating a clear research gap. As such, we expect that the impact of BCT is highly likely to be amplified to improve SC performance in high-speed markets. Our argument is dependent on existing research that demonstrates how knowledge dissemination might result in increased variance in performance results in turbulent environments [34]. Hence, it can be contended that ED generates pressure on companies to utilize organizational knowledge as a guideline for decision-making. Consequently, there is a need for a more profound comprehension of the relationship between BCT, SCI, and SCR, as well as how ED moderates the BCT-SCI relationship.

This study makes a valuable contribution to the literature on operations management, information systems management and strategic management regarding the role of BCT in enhancing SCI to improve SCR under the influence of ED. Specifically, this study contributes in several ways. First, it examined the effect of BCT on SCR. This was a response to a call made by Sheel and Nath [35] to probe the impact of BCT adoption on critical SC performance parameters such as resilience. Second, it explored the mediating role of SCI between BCT and SCR. Third, it highlighted that ED can condition the extent to which BCT can enhance SCI and thus SCR. Finally, it examined the proposed model in the context of the automotive industry in India.

## 2 Theoretical background and hypotheses derivation

### 2.1 Theoretical basis

Relying on the DCT [e.g., 36] and the CT [e.g., 37], the conceptual model that illustrates the relationships among the key constructs of this study has been developed (see Fig 1).

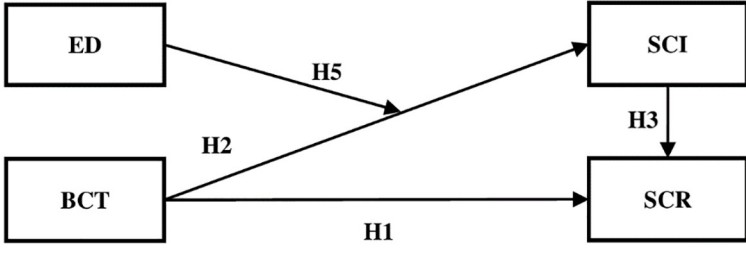

**Fig 1. Research model.** Source: Authors' own work.

According to DCT, a company seeking for long-term competitive advantage must either develop new resources and capabilities or deploy existing resources and capabilities to deal with emerging chances [38]. A DC is "the firm's ability to integrate, build, and reconfigure internal and external competences to address rapidly changing environments [36]." DCT is an expansion of the Resource-Based View (RBV) that elucidates how enterprises can attain a competitive edge in turbulent environments [18, 39]. There are numerous forms of capabilities, spanning from basic functional capabilities to dynamic high-level capabilities that are critical to an enterprise's strategic success [40]. DCs are critical for business survival, especially in quickly changing environments, such as those presently confronting manufacturing enterprises as a result of shifting market structures and technologies [41].

DCT has been widely adopted in the literature of operations management [42, 43]. The results demonstrate that DCs can be produced inside a focal firm in partnership with external partners within the SC, involving the reconfiguration of operating procedures to increase effectiveness. In SCM, this process entails developing the capabilities necessary to respond properly to changing environmental and market conditions [40]. Prior studies have identified SCR as a DC for anticipating and recovering from unavoidable risk events [10, 44]. SCR, as a DC, helps organizations to absorb the unfavorable consequences of a variety of risk sources [45]. As such, the ability of the SCR to absorb unforeseen interruptions and restore the SC to its former or improved case may result in competitive advantages [46]. Similarly, BCT capability is visualized in previous studies as a DC [e.g., 17]. As a result, we employ DCT as the theoretical foundation for the current study [36]. We theorize that BCT is a DC that can offer a competitive edge to a firm. Implementation of BCT can help firms to reduce the level of risk, as such the risk of information distortion across the SCs [47] by increasing transparency, accountability and visibility in SCs [48]. Therefore, we argue that BCT improves SCR via the intermediating role of SCI.

Although DCT is widely adopted, several researchers have argued that DCT is context-insensitive [49, 50]. The impact of DCs on an enterprise's potential to attain better performance depends on the context in which the firm acts [36]. Consequently, we suggest that it is vital to analyze the circumstances under which capabilities are most valued. Contingency theory (CT) addresses this idea of context's importance in elucidating how a firm's inner and outer conditions result in disparate performance results [18, 51]. Thus, managers must perform an in-depth analysis of the organization's environment, taking into account internal firm features, and change practices accordingly [52]. CT has been identified as a critical theoretical lens for understanding the contextual conditions in which efficient operations management methods can be implemented [53], which contributes to the theoretical precision of research [54]. Hence, while considering CT, a variety of concepts of fit can be used and should be explicitly evaluated during the research process [53]. As a result of Schilke's [55] work, we adopt a contingency viewpoint operationalized through a fit moderation notion, which argues that the differential effects of BCT on SCI are dependent on the degree of the moderating variable (in this case ED). Since the objective of DCs is to equip organizations with the capacity to adapt to rapidly changing environments [56], we integrate DCT and CT to build the theoretical foundation of this study.

## 2.2 Blockchain technology and supply chain resilience

BCT is a beneficial tool to enhance resilience in contemporary SCs operating in a more dynamic business environment [57]. Previous research has revealed the positive influence of BCT on SC performance in general and on SCR in particular [e.g., 19, 58, 59]. According to Bayramova et al. [60], BCT has consequences for SCR in terms of visibility, information

sharing, risk management, and integration. The visibility metric is typically enhanced when BCT are adopted in the form of traceability systems [61], whereas information exchange and collaboration are typically enhanced when BCT are implemented as distributed ledger technology features [62, 63]. BCT is well-suited to service clients by allowing the tracking and tracing of orders from production to delivery and adjusting promptly [64]. BCT enables enhanced visibility of SCs and network-wide real-time data sharing. As a consequence, it can aid SCR strategies by minimizing the number of stakeholders impacted by a disruption [65]. In this context, Lambourdiere and Corbin [48] indicate that firms must incorporate BCT into the logistical processes of their SCs. By doing so, they can utilize this technology to develop capabilities within the SCs, which ultimately leads to the creation of more robust and resilient SCs. On the basis of the foregoing, it can be assumed that:

**H1:** BCT is significantly and positively associated with SCR.

## 2.3 Blockchain technology, supply chain integration, and supply chain resilience

BCT is digitally predisposed to integrate all SC processes among partners [35, 66], with many advantages such integration enables, such as "product traceability, settlement of transactions, process automation, and execution of smart contracts" [64]. According to Polim et al. [67], one of the key capabilities of BCT is the integration of information. SCI, which is enabled through BCT, is highly secured, as BCT prevents unauthorized access to the information stored on the ledger [17]. BCT facilitates the integration of supplier and customer information, resulting in exceedingly high levels of SCI [68]. BCT accelerates the execution of business activities while maintaining a high level of reliability and accuracy [69]. BCT allows records to be shared with SC partners [70, 71], hence resolving trust difficulties among partners [72]. Each member partner has access to the other's internal procedures. Kshetri [47] proposes incorporating BCT with the Internet of Things to determine the source of disturbances in SC and to effectively address crises. This incorporation enables the reduction of uncertainties and promotes enhanced process integration [73] and SC transparency [27]. Moreover, the incorporation of BCT in SCs leads to enhanced privacy, audibility, and increased operational efficiency [17]. Hence, we suppose that:

**H2:** BCT is significantly and positively associated with SCI.

Moreover, IT-based SCI allows sharing of data or information in real time [74]. Internal integration, as highlighted by Tiwari [75], facilitates the integration of all internal functions within an organization, resulting in improved communication and efficient decision-making processes [76]. By enabling the sharing of information, internal integration plays a crucial role [77], on the other side, operational integration between SC partners enhances SCR in response to disturbances [74]. Related to this, a study conducted in Taiwan by Liu and Lee [78] demonstrated that both internal integration and customer integration, which are forms of SCI, have a significant positive impact on SCR, particularly within third-party logistics providers. Furthermore, supplier integration can have a positive influence on enhancing SCR in terms of effectively dealing with uncertainties and responding promptly to disruptions in the SC [79]. Long-standing partnerships within the SC with suppliers who exhibit increasing levels of innovativeness can significantly impact SCR, as the pivot company has strong partnering relationships that enable it to quickly step back when faced with disruptions [80]. SCI improves SCR to build SC partnerships [2]. Firms' information technology can incorporate the system to enhance its response as a form of SCR [78]. Recent studies examining the impact of ICT-

enabled integration technologies like SC information systems on SC performance [e.g., 17], highlight SCI as a crucial mediating variable. Digital technologies-enabled SCI has a twofold effect on information processing demand and capacity. Additionally, digital technologies-enabled internal integration brings synergistic advantages, which improve the capability to manage the flow of information. This allows firms to swiftly prevent and respond to disruptions, thereby enhancing their resilience [81]. Based upon the preceding arguments, we suppose that:

**H3:** SCI is significantly and positively associated with SCR.

**H4:** SCI mediates the association of BCT with SCR.

## 2.4 The moderating role of environmental dynamism

ED is defined as "the volatility and unpredictability of the firm's external environment [55]." It is a critical factor in DCT [55], implying that the differential influences of DC on SC characteristics [82] and organizational performance [34] are dependent on the external environment's dynamism [38]. According to Eisenhardt and Martin [38], firms generally follow predictable and linear pathways in moderately dynamic marketplaces (characterized by defined market boundaries and stable industry structures). Therefore, effective DC in moderately dynamic environments is contingent upon making use of present knowledge. In comparison, changes in fast-moving markets (characterized by complex and ambiguous structures) are typically nonlinear and unpredictable [30]. In such dynamic environments, firms within SCs increasingly rely on collaboration and integration among stakeholders to adapt to changes and disruptions. Therefore, the role of new technological applications such as BCT in SCs becomes more important [57, 83]. Changes in the more complex and turbulent environment are driving firms to adopt BCT [84]. According to Meidute-Kavaliauskiene et al. [57], firms need to invest in BCT to respond swiftly to market changes and consumer expectations in today's volatile business environment. BCT is an important technology for firms to better control the flow of the SC. Furthermore, Liu and Li [85] asserted that BCT is well-suited to unpredictable and frequently changing environments and laws. In contrast, Orji et al. [86] claim that market dynamism has a negligible impact on BCT adoption in the freight logistics business and ranking fourth in the institutional context. Indeed, Meidute-Kavaliauskiene et al. [57] claim that the unforeseeable nature of ED's effect on organizational results gives companies with additional chances to leverage and explore BCT capabilities. Therefore, a new requirement to respond to market changes is to assess the utilize of digital technology in SC processes within the context of a dynamic market environment.

Environmental turbulence which entails uncertainty can have an effect on the adoption of BCT [87]. However, Wamba et al. [30] found that the effect of big data analytic on agility and adaptability did not differ under the influence of ED. In contrast, Liu et al.'s [88] study revealed that ED moderates the indirect influence of digital technologies such as BCT on environmental and economic performance via digital SC platforms. Thus, in order to achieve better performance, manufacturing organization must not only leverage on inner information processing abilities that are enabled by digital technologies, but also leverage the most advanced digital SC platforms to get additional information outside, particularly in a dynamic context. On the basis of the foregoing, we argue that ED can enhance the impact of BCT on SCI, hence affecting SCR. Accordingly, we propose the following:

**H5:** ED positively moderates the association of BCT with SCI.

## 3 Methodology

### 3.1 Sampling and data collection

In this study, the proposed model depicted in Fig 1 was evaluated through a survey-based methodology with data collected from Indian firms in the automotive industry. The firms were chosen from the "Society of Indian Automotive Manufacturers" and the "Automotive Components Manufacturers Association of India" databases. The questionnaire was designed and mailed to 300 managers from 100 firms. The authors took the help of a private market research firm to administer the questionnaire and collect the data. Participation in the survey was completely optional and restricted to only those respondents who had worked in the automotive industry for at least two years. This was done to ensure that some respondents had some level of acquaintance with the industry. In addition, only participants with prior knowledge of BCT and SCM concepts were asked to complete the survey. The target respondents were the supply chain/ logistics/production/manufacturing/digitalization and technology lead managers. To incentivize participants to fill out the questionnaire and boost the response rate, the questionnaire included a statement guaranteeing respondents' anonymity. Furthermore, regular e-mail reminders were sent out. The total number of respondents in this study was 300. Of this number, 148 were returned while 141 were complete and useable, representing a response rate of 47%. This is a good percentage in comparison to those mentioned in earlier research [e.g., 89]. Following these procedures, data collection took three months (from mid-June to mid-September 2022).

This study's participants varied in terms of gender (Male-84.4% and Female-15.6%); educational level (Secondary and below-13.48%, Undergraduate-60.99%, and Postgraduate-25.53%); work experience (Less than 15 years-22.7% and 15 years and above-77.3%); age (20 to 29years-3.55%, 30 to 39years-14.9%, 40 to 49years-52.48%, and 50 years and above-29.07%); and position (logistics manager-45.39%, production/manufacturing manager-34.04%, and digital technology/ICT manager-20.57%).

### 3.2 Measures

For this study, the survey questionnaire instrument was used to collect the required data to examine the links in the proposed model. Initially, we conducted 6 personal interviews with academics and business professionals in order to ensure that the proposed survey questions are understandable and not ambiguous, vague, or difficult to reply [90]. The constructs and corresponding items employed for their measurement can be found in S1 Appendix.

To assure reliability and construct validity, all measurement items were obtained from existing literature and adapted to be appropriate to the context of this study. SCR was measured using 4 items derived from Al-Hakimi *et al*. [11]. For SCI, it was measured using 10 items derived from Kamble *et al*. [17]. While BCT was measured using 9 items derived from Kamble *et al*. [17] and Dubey *et al*. [18] and lastly ED was measured using 3 items derived from Wamba *et al*. [30].

### 3.3 Common method bias (CMB)

After the data collection phase, the initial step involved conducting a test for common method bias (CMB) before proceeding with further statistical analysis using the gathered data. CMB is "a common issue in statistical-based investigations when the data is collected from a single respondent from a firm, which may lead to an artificial increase in sample sizes and inflated estimates" [91]. To mitigate CMB, several steps were taken, including ensuring the clarity of measurement items, anonymizing participants, and selecting participants who possessed

knowledge of BCT and SC management [ibid]. Besides that, "Harman's one-factor" test was carried out as per the procedures of Podsakoff et al. [92], in order to verify the absence of CMB. According to Podsakoff et al. [92], a preliminary factor analysis is conducted for all questionnaire items such that if a single factor stands out in the analysis or if the first factor elucidates over 50% of the variance, it indicates a substantial effect of error variance. While previous studies have indicated that Harman's method might not effectively identify CMB in comparison to other tests, some recent studies have confirmed that it is a very beneficial method [e.g., 93]. In our study, the factor identified accounted for 35% of the total variance, indicating that CMB was not a concern for the collected data.

Furthermore, an alternative approach proposed by Fuller et al. [94] involved examining the collinearity variance inflation factor (VIF) using SmartPLS to assess the presence of CMB. The findings from this analysis indicated that the VIF values were below the recommended threshold of 3, as proposed by Fuller et al. [94]. Hence, the data does not raise concerns regarding the presence of CMB.

## 4 Data analysis and results

To analyze the relatively intricate model in this study, we utilized the PLS-SEM approach via SmartPLS 4 software, following the guidelines outlined by Ringle et al. [95]. The widespread utilization of PLS-SEM in administrative studies can be attributed to its numerous advantages [96]. Specifically, when working with smaller sample sizes [97] and when the research primarily focuses on prediction [93]; where it exhibits greater statistical power compared to "covariance-based SEM" (CB-SEM) when adopted with complex models with limited sample sizes [ibid].

Nevertheless, in recent times, some researchers have expressed concerns regarding the alleged inappropriate application of PLS-SEM, particularly concerning the arguments supporting its use in scenarios involving "small sample sizes, large model complexity, less restrictive distributional assumptions, and less restrictive utilization of formative measurement models" [93]. For instance, Evermann and Rönkkö [98] have raised a few questionable arguments, notably claiming that PLS-SEM is "a well-known bias estimator", often indicated to as the "PLS-SEM bias". However, simulation experiments have shown that the discrepancies between the estimations of PLS-SEM and CB-SEM are minimal [99]. Consequently, the widely examined PLS-SEM bias has little influence on the results of practical applications because of the asymptotic accuracy of estimates under consistent large-scale assumptions (e.g., a large sample size and a substantial number of indicators per latent variable) [100].

Furthermore, although PLS-SEM tends to produce biased estimates on average, these estimates exhibit lower variability when compared to the estimates resulting from CB-SEM [99]. This characteristic proves advantageous in research contexts where CB-SEM, based on maximum probability, often yields inflated standard errors [101] and violates certain assumptions, including "high model complexity, small sample size, non-normal data". The enhanced efficiency in parameter estimation is evidenced by the greater statistical power of PLS-SEM in comparison to CB-SEM. This aligns with the current analysis, as PLS-SEM is well-suited for examining relationships between multiple constructs simultaneously, even with a sample size of 141 cases. Overall, the PLS model encompasses two interdependent models: the "measurement model" and the "structural model".

### 4.1 Measurement model

For this study, the confirmatory composite analysis (CCA) approach, as outlined by Hair et al. [102], was employed to assess the measurement model. To ensure reliability, the values of

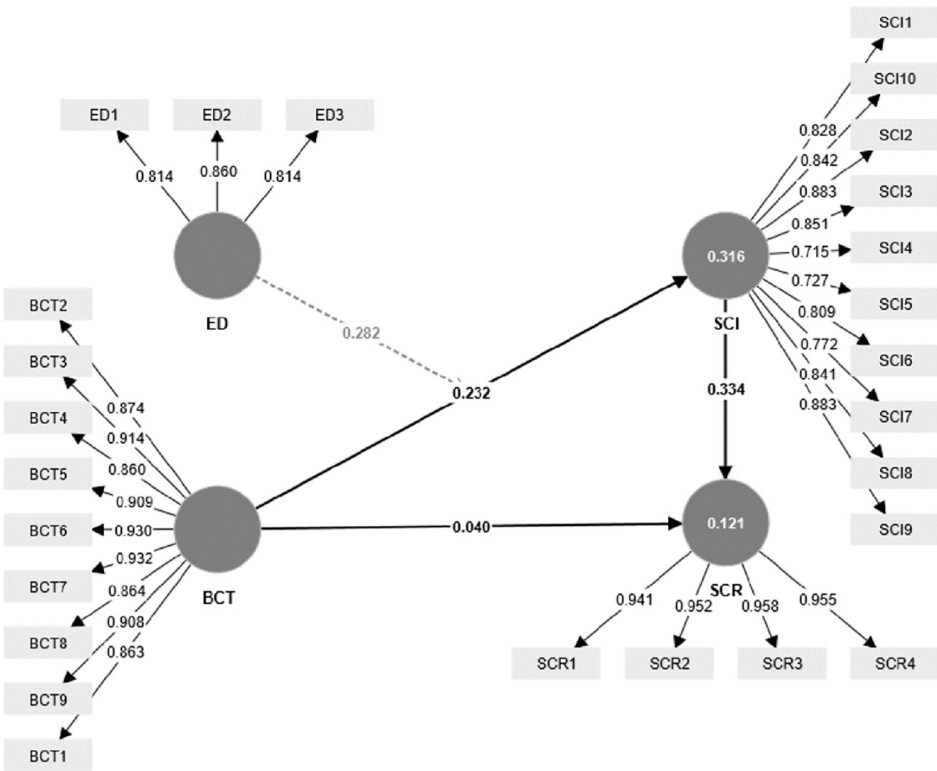

**Fig 2. Measurement model.** Source: Authors' own work based on the statistical analysis (Smart PLS).

"Cronbach's alpha (α)" and "composite reliability" (CR) needed to surpass 0.70, in accordance with Nunnally and Bernstein [103]. Furthermore, the construct validity was evaluated through the examination of "convergent validity" and "discriminant validity", following the guidelines provided by Hair et al. [102]. Convergent validity was validated when the value of "average variance extracted (AVE)" for each construct exceeded 0.50, as suggested by Hair et al. [104]. Additionally, the factor loadings for each item depicted in Fig 2 and Table 1 were required to exceed a minimum of 0.70 [ibid].

Moreover, the "Heterotrait-Monotrait (HTMT)" method, as outlined by Henseler et al. [105], was employed to verify discriminant validity. According to Al-Swidi et al. [93], the values within the HTMT matrix, especially between the constructs, must not surpass 0.90. In our study, the results demonstrated that the values did not surpass this threshold, as presented in Table 2.

From the results presented in Tables 1 and 2, it is evident that all requirements, including loadings, reliability, and validity, were met, which emphasizes the measurement model validity.

## 4.2 Structural model

Following the guidelines of the second step of the CCA approach, the structural model was evaluated in this study, as outlined by Hair et al. [102]. The significance of the paths in the model (as shown in Fig 3) was assessed using t-statistics, calculated through a bootstrapping technique [106].

**Table 1. Loadings, reliability, and convergent validity.**

| Constructs | Items code | Factor loading | CR (α) | AVE | Convergent validity |
|---|---|---|---|---|---|
| BCT | BCT1 | 0.863 | 0.975 (0.969) | 0.802 | Yes |
| | BCT2 | 0.874 | | | |
| | BCT3 | 0.914 | | | |
| | BCT4 | 0.860 | | | |
| | BCT5 | 0.909 | | | |
| | BCT6 | 0.930 | | | |
| | BCT7 | 0.932 | | | |
| | BCT8 | 0.864 | | | |
| | BCT9 | 0.908 | | | |
| SCI | SCI1 | 0.828 | 0.949 (0.945) | 0.667 | Yes |
| | SCI2 | 0.883 | | | |
| | SCI3 | 0.851 | | | |
| | SCI4 | 0.715 | | | |
| | SCI5 | 0.727 | | | |
| | SCI6 | 0.809 | | | |
| | SCI7 | 0.772 | | | |
| | SCI8 | 0.841 | | | |
| | SCI9 | 0.883 | | | |
| | SCI10 | 0.842 | | | |
| ED | ED1 | 0.814 | 0.782 (0.775) | 0.688 | Yes |
| | ED2 | 0.860 | | | |
| | ED3 | 0.814 | | | |
| SCR | SCR1 | 0.941 | 0.968 (0.965) | 0.905 | Yes |
| | SCR2 | 0.952 | | | |
| | SCR3 | 0.958 | | | |
| | SCR4 | 0.955 | | | |

Table 3 shows the results of the hypotheses testing. The results demonstrate that paths (BCT→SCI) (β = 0.232, p<0.01) and (SCI→SCR) (β = 0.334, p<0.01) were positive and significant, supporting H2 and H3. In addition, the path (BCT→SCR) (β = 0.040, p>0.05) was insignificant when there was no putative mediator (SCI); however, this effect of BCT on SCR became significant (β = 0.077, p<0.05) when the putative median was included. Therefore, hypothesis H1 is not supported.

Along with the linear paths of our proposed model, we investigated the moderating effect of ED on the path linking BCT and SCI, as the results revealed that ED positively and significantly moderates the path (BCT→SCI) (β = 0.282, p<0.01). Hence, hypothesis H5 is supported (see Fig 4). The outcome reveals that a strong correlation exists between high ED and increased levels of SCI, particularly when companies implement a high degree of BCT.

**Table 2. Discriminant validity.**

| Constructs | BCT | SCI | ED | SCR |
|---|---|---|---|---|
| BCT | | | | |
| SCI | 0.300 | | | |
| ED | 0.190 | 0.428 | | |
| SCR | 0.146 | 0.361 | 0.105 | |

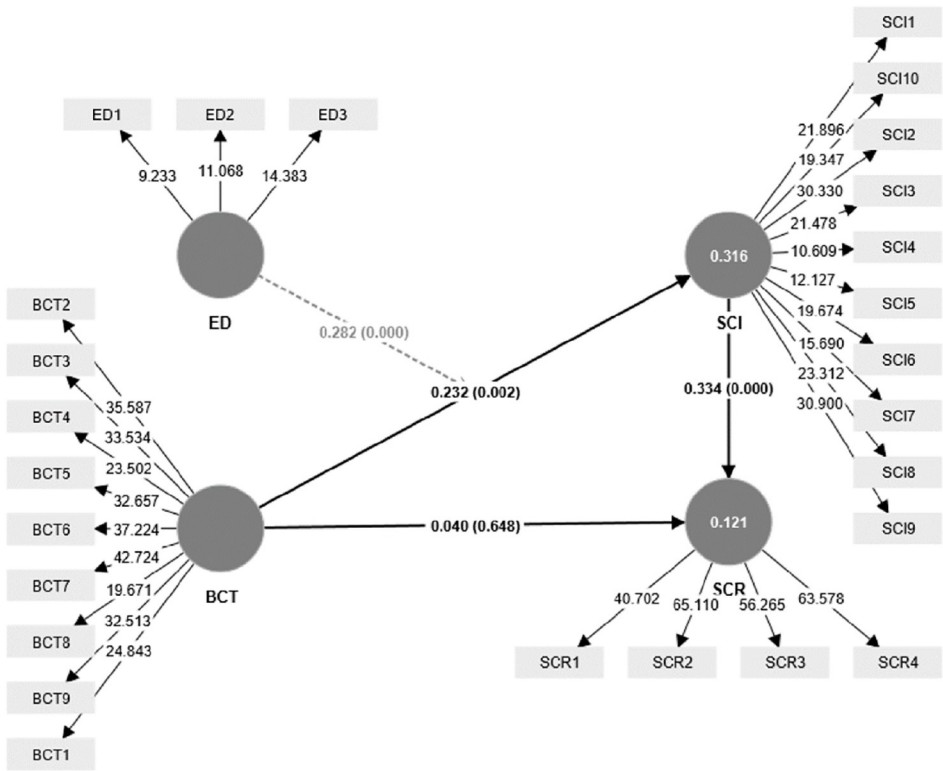

**Fig 3. Structural model.** Source: Authors' own work based on the statistical analysis (Smart PLS).

Moreover, in accordance with the guidelines presented by Sarstedt et al. [107], the mediating role of SCI between BCT and SCR was investigated. The findings, which are outlined in Table 4, provide confirmation that SCI acts as a full mediator in the BCT-SCR relationship. Thus, H4 is supported.

As a next step, the explanatory power of the study model was evaluated by calculating the explained variance ($R^2$) of the endogenous constructs, where the $R^2$ values in the model were as follows: SCI (0.316) and SCR (0.121), as demonstrated in Table 5. To assess the results, Chin's guidelines for prediction "0.10 = weak, 0.33 = moderate, 0.67 = large" were utilized [108].

Additionally, Cohen's $f^2$ guidelines were employed to evaluate the effect size of each predictor [109], where the $f^2$ values of 0.35, 0.15, and 0.02 are classified as 'large', 'medium', and 'small', respectively. Accordingly, the effect size of BCT on SCI was 0.076; BCT on SCR was 0.002; ED on SCI was 0.242; and SCI on SCR was 0.114 (see Table 5). Moreover, the predictive capability of the model was assessed through Stone-Geisser ($Q^2$). The $Q^2$ values for the

**Table 3. Direct and moderation effect.**

| Direct paths | β | t value | p value | Decision |
|---|---|---|---|---|
| BCT→SCR | 0.040 | 0.457 | 0.648 | *Not supported* |
| BCT→SCI | 0.232 | 3.142 | 0.002 | *Supported* |
| SCI→SCR | 0.334 | 3.751 | 0.000 | *Supported* |
| Moderation | β | t value | p value | Decision |
| ED*BCT→SCI | 0.282 | 3.947 | 0.000 | *Supported* |

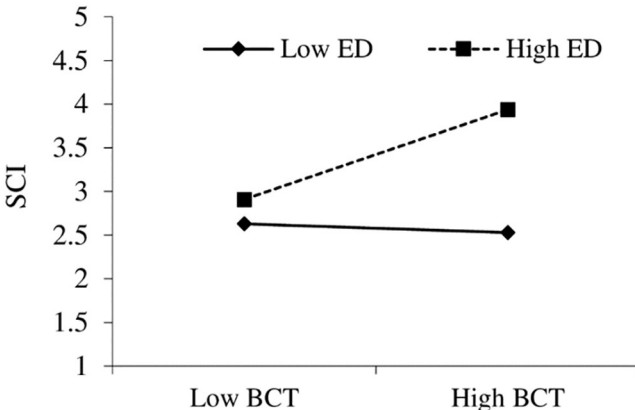

**Fig 4. Moderation effect of ED on BCT and SCI.** Source: Authors' own work based on the statistical analysis (Smart PLS).

endogenous constructs, SCI and SCR, were 0.264 and 0.040, respectively. These values, being above zero, indicate satisfactory predictive relevance [106].

As a last evaluation of the structural model's predictive abilities, the PLSpredict procedure was executed to assess the prediction errors, following the methodology outlined by Manley et al. [110]. The evaluation process involved the calculation of $Q^2$ and a comparison of the prediction errors between PLS and LM. Table 6 presents the $Q^2$ values obtained by comparing the prediction errors of the PLS results with those of the mean predictions. All $Q^2$ values were found to be higher than zero, which indicates that the prediction error associated with the PLS results was less than the prediction error resulting from depending solely on mean values. Moreover, the differences between LM and PLS in terms of indicators such as "mean absolute error (MAE)" and "root-mean-square error (RMSE)" were relatively minor. As per the recommendations provided by Hair et al. [102], "the model has medium predictive power when only a few indicators in the PLS analysis exhibit larger prediction errors compared to the LM criterion". Therefore, the model's predictive validity was verified.

## 5 Discussion

Guided by DCT and CT, we explored how and when BCT adoption improves SCR. According to the results, BCT does not directly affect SCR. This result is in line with the previous studies [e.g., 111], which revealed that digital technologies (In this case, BCT) had an insignificant effect on SCR in the presence of putative mediators. On the contrary, it contradicts the findings of Min [19], which have previously demonstrated a positive effect of BCT on SCR. The results also indicate that BCT positively affects SCI. This is in line with the results of prior research [e.g., 17]. On the other hand, our results reveal that SCI positively affects SCR, which is similar to the results of Siagian et al. [74] and Tarigan et al.'s [79] studies.

**Table 4. Indirect effect.**

| Mediation paths | Indirect path | | | Direct path | | | Decision |
|---|---|---|---|---|---|---|---|
| | β | t value | p value | β | t value | p value | |
| BCT →SCI →SCR | 0.077 | 2.445 | 0.015 | 0.040 | 0.457 | 0.648 | *Fully mediation* |

**Table 5. $R^2$, prediction, and effect size.**

| Constructs | $R^2$ | $Q^2$ | $f^2$ in relation to | | |
|---|---|---|---|---|---|
| | | | SCI | | SCR |
| BCT | | | 0.076 | | 0.002 |
| ED | | | 0.242 | | |
| SCI | 0.316 | 0.264 | | | 0.114 |
| SCR | 0.121 | 0.040 | | | |

As expected, the findings also show that SCI mediates between BCT and SCR, where BCT indirectly affects SCR through the full mediation of SCI. Importantly, the results show that the association between BCT and SCI will be stronger under a high level of ED.

As a result, this study contains some noteworthy contributions to theory and evidence for managers, as detailed in the following sections.

## 5.1 Theoretical implications

The theoretical contribution of the current study is many folds. First, based on DCT and CT, this study contributes to theoretical arguments surrounding DCs (in this case, BCT and SCR) mediated by SCI and under the conditional impacts of ED. The current study can be considered an attempt to incorporate literature from three areas: operations management, information systems management, and strategic management. While Dubey et al. [112] and Wamba et al. [30] have attempted to fill the divide between operations management and information systems literature in the past, these studies have relied on DCT or information processing theory, or the incorporation of institutional theory and RBV. Second, the results of the study pertinent to the role of BCT for SCR complement the previous studies on other SCR enablers such as IT [113], as technical sources for SCR. In this regard, a firm's capability to integrate a BCT into its overall operating structure is as a barometer of its capability to develop SCR. Furthermore, we respond to a call made by Ying et al. [114] to enrich the present status of exploratory research on BCT by providing more empirical evidence. In fact, the majority of the

**Table 6. PLSpredict assessment.**

| Indicators | $Q^2$ | PLS | | LM | |
|---|---|---|---|---|---|
| | | RMSE | MAE | RMSE | MAE |
| SCI1 | 0.067 | 0.644 | 0.508 | 0.632 | 0.503 |
| SCI2 | 0.086 | 0.707 | 0.526 | 0.684 | 0.518 |
| SCI3 | 0.080 | 0.695 | 0.531 | 0.727 | 0.559 |
| SCI4 | 0.258 | 0.533 | 0.369 | 0.525 | 0.390 |
| SCI5 | 0.270 | 0.609 | 0.421 | 0.603 | 0.453 |
| SCI6 | 0.300 | 0.548 | 0.434 | 0.607 | 0.473 |
| SCI7 | 0.233 | 0.580 | 0.453 | 0.662 | 0.519 |
| SCI8 | 0.067 | 0.635 | 0.503 | 0.630 | 0.504 |
| SCI9 | 0.072 | 0.723 | 0.544 | 0.730 | 0.553 |
| SCI10 | 0.061 | 0.676 | 0.529 | 0.720 | 0.571 |
| SCR1 | 0.030 | 0.574 | 0.524 | 0.618 | 0.554 |
| SCR2 | 0.041 | 0.604 | 0.541 | 0.661 | 0.569 |
| SCR3 | 0.024 | 0.575 | 0.525 | 0.617 | 0.552 |
| SCR4 | 0.048 | 0.592 | 0.536 | 0.650 | 0.568 |

existing literature on BCT is a review of the literature [e.g., 115, 116] and is conceptual in nature [e.g., 117, 118]. While a few researchers have made efforts to study BCT empirically, their studies were qualitative in nature [87], relatively narrow in scope [114], and based on the Technology Acceptance Model [68] or Unified theoretical frameworks of the Acceptance Model [119]. Thus, by combining the theoretical lens of the DCT and CT frameworks with empirical evidence from Indian automotive manufacturing firms, this study contributes to the expanding body of knowledge on BCT and diversify the literature on operations management and information technology. Third, this study examines the mediating role of SCI between BCT and SCR. In comparison to the present research on the importance of IT, our results propose that the relationship between BCT and SCR is more likely to be complex, not narrow-scoped. In doing so, we add to the work of Pattanayak et al. [120] by proposing the mechanism through which BCT improves SCR. Fourth, this study establishes that ED plays a crucial role in the link between BCT and SCI. Thus, our study contributes to bridging the critical research gap concerning the effect of BCT on SCI and under what conditions do BCT contribute to the enhancement of DCs (SCP), which are probably among the most urgent research problems in the domain of operations and SCM. The inclusion of ED into the image develops a conditioned vision of the function that BCT plays in SCR indirectly via SCI. By incorporating existent arguments on DCs under varied levels of ED, we argue that DCT [36] is insufficient for dealing with highly dynamic and uncertain environments. As a result, our results imply that the interaction between BCT, SCI, and SCR may be more complicated than a simpler linear relationship.

## 5.2 Managerial implications

The study's findings have valuable managerial implications for firms seeking to develop SCR and attain high performance in turbulent environments. First, an in-depth analysis of the findings shows that firms should adopt BCT to minimize the influences of SC disruptions and enhance SCR as a whole. The study's findings imply that managers should carefully assess the SC capabilities (SCR) for sensing dynamic changes in the inner and outer environment, which may aid in shaping chances and mitigating disturbances. In general, managers realize the issues surrounding SC disruptions and the critical role of SCR [121]. Nevertheless, their knowledge of the role of BCT in developing SCR may be less complete. Furthermore, the current study emphasizes and reminds managers of the crucial role of resilience in managing SC disruption as it occurred in the unique period of the COVID-19 pandemic and in the event of future epidemic outbreaks or crises. This highlights the critical and pressing need for enterprises, as well as governments, to invest in developing fundamental capabilities of resilient SCs in order to improve performance during such crises. Although SCR is a relatively new term that is not widely discussed in some developing countries, many firms in India for example are beginning to realize that improving SCR is critical to their global competitiveness, which requires greater knowledge about risk mitigation strategies [122]. Second, our findings indicate that BCT is a capability that does not impact a firm's performance outcomes [e.g., 123], but is also utilized for the development of other capabilities, such as SCR. However, this effect is indirect, as our findings inform practice by indicating the sequence in which capabilities should be developed and implemented (BCT→SCI→SCR). Pursuing to embrace the BCT-enabled DCs in SCs aids in developing a firm's response to disturbances in the form of SCR, which eventually boosts SC performance [122]. Our findings indicate that adopting BCT enhances SCI, where it dramatically minimizes the number of partners influenced by perturbation, the cost of the disruption, and the time required for the network to recover. Adopting BCT can be a fruitful solution for enhancing SCR, however, the costs and time it takes to set

up and run must be taken into account. In this regard, managers should adopt BCT as a solution within the context of stated joint risk management systems within the SC, which should include a joint risk strategy. However, it is necessary to upgrade and connect the underlying risk management procedures before implementing BCT solutions. Furthermore, managers interested in adopting and developing BCT-based IT solutions for the SC should begin lobbying SC partners to design a regulatory framework for BCT. Without a regulatory framework, the technology will remain extremely dangerous to adopt. At the moment, in a country like India, there is no specific legal structure in place to address BCT [35]. India has begun to investigate BCT implementation in the SC [68], and India now has an opportunity to improve its global competitiveness [124]. However, given the additional transparency brought about by BCT adoption, corporations are recommended to conduct a thorough study to ascertain partners' reactions to a fully transparent SC that allows for close monitoring of customers and other parties, including rivals [117]. Third, our work demonstrates the importance of DCs in developing countries. Numerous earlier studies have established that DCs are only useful in a fast-paced context. Our findings indicate that DCs are important in both developed and developing countries. This is because dynamism encompasses more than competition and rapid invention. Another factor to consider is susceptibility to interruptions. This is a more prevalent or serious issue in developing countries. As a result, enterprises in developing nations must invest in creating DCs if they seek to survive in the face of continual disturbances. While some routines emerge by chance, others require managers to have patience and forethought in determining when and how to construct DCs, as well as how to explore and use DCs concurrently to achieve a competitive edge.

## 5.3 Limitations and future research

Like other studies, this study has some limitations. First, while the DCT has garnered considerable attention, we contend that it suffers from context insensitivity, as stated by Ling-Yee [49]. We explicate context sensitivity to mean that DCT is incapable of identifying the circumstances in which DCs are most helpful [55, 112]. Although we tried to deal with this limitation of the firm's DCT by incorporating DCT with CT, when examining the moderating role of ED, we believe that further research can be conducted to determine the optimal conditions under which BCT can achieve SCI in order to improve SCR. Second, as with any survey-based study, this one has limitations, such as CMB or endogeneity [125]. As a result, we used statistical approaches to detect CMB. However, this study cannot rule out the potential of CMB. Future study may employ longitudinal or multiple informant data to examine the potential links in the research model. Third, the results of this study depend on managers' perspectives in the context of the automotive industry, so they are not generalizable in the context of services that also suffer from disruptions in SCs. Finally, we examined ED as a proxy for market dynamism in our study. Considering the significance of this concept in the context of DCs, additional sources of dynamism may be explored, including uncertainty, competitiveness, and technology.

## Supporting information

**S1 Appendix. Questionnaire items.**
(DOCX)

**S1 Dataset.**
(RAR)

## Author Contributions

**Conceptualization:** Abdullah Kaid Al-Swidi, Mohammed A. Al-Hakimi.

**Data curation:** Mohammed A. Al-Hakimi.

**Formal analysis:** Abdullah Kaid Al-Swidi.

**Investigation:** Abdullah Kaid Al-Swidi, Hussam Al Halbusi.

**Methodology:** Abdullah Kaid Al-Swidi, Mohammed A. Al-Hakimi.

**Resources:** Abdullah Kaid Al-Swidi, Hussam Al Halbusi, Jaithen Abdullah Al Harbi.

**Supervision:** Abdullah Kaid Al-Swidi, Jaithen Abdullah Al Harbi.

**Validation:** Abdullah Kaid Al-Swidi, Mohammed A. Al-Hakimi, Hussam Al Halbusi.

**Visualization:** Mohammed A. Al-Hakimi.

**Writing – original draft:** Abdullah Kaid Al-Swidi, Mohammed A. Al-Hakimi.

**Writing – review & editing:** Abdullah Kaid Al-Swidi, Hussam Al Halbusi, Jaithen Abdullah Al Harbi, Hamood Mohammed Al-Hattami.

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
