## [Decision Letter · Decision Letter 0]

25 Jul 2023

PONE-D-23-18950Does blockchain technology matter for supply chain resilience in dynamic environments? The role of supply chain integrationPLOS ONE

Dear Dr. Al-Hakimi,

Thank you for submitting your manuscript to PLOS ONE. After careful consideration, we feel that it has merit but does not fully meet PLOS ONE’s publication criteria as it currently stands. Therefore, we invite you to submit a revised version of the manuscript that addresses the points raised during the review process.

We look forward to receiving your revised manuscript.

Kind regards,

Kittisak Jermsittiparsert, Ph.D.

Academic Editor

PLOS ONE

Journal Requirements:

Additional Editor Comments:

Please revise the paper as reviewers' suggestions.

Reviewers' comments:

Reviewer's Responses to Questions

**Comments to the Author**

1. Is the manuscript technically sound, and do the data support the conclusions?

Reviewer #1: Yes

Reviewer #2: Yes

2. Has the statistical analysis been performed appropriately and rigorously? 

Reviewer #1: Yes

Reviewer #2: Yes

3. Have the authors made all data underlying the findings in their manuscript fully available?

Reviewer #1: Yes

Reviewer #2: Yes

4. Is the manuscript presented in an intelligible fashion and written in standard English?

Reviewer #1: Yes

Reviewer #2: Yes

5. Review Comments to the Author

Reviewer #1: - The sentence "By enabling the sharing of information, internal integration plays a crucial role [77]" (before hypothesizing H3) has too old citation (2012) since the up-to-date paper prefers the citation between 2019-2023. You may replace this citation "Aunyawong, W., Wararatchai, P., & Hotrawaisaya, C. (2020). The influence of supply chain integration on supply chain performance of auto-parts manufacturers in Thailand: a mediation approach. International Journal of Supply Chain Management, 9(3), 578-590." because this paper depict the effects of supply chain integration on resilience (flexibility), especially in automotive industry.

- In 3.1 Sampling and Data Collection, The word "logistic manager" is incorrect, "logistics manager" is correct.

- In 3.2 Measures, the numbers of items should be specified. (for example, 4 items of SCR were derived from...) and use "were" because each variable has several items.

- 4.1 *M*easurement model (start with the capital letter)

- The name of Table should be changed to "Direct and Moderation Effect" to suit with the info. in the table.

- Under Table 5, correct how to write Q-Square in Line 3 (Q2 is incorrect since it can lead to misunderstand as Second Quartile)

- The name of Figure 4 "Interactive effect of BCT and ED on SCI" is incorrect because the figure explains the moderation effect of ED on the path linking BCT and SCI. (Suggested Figure Name: "Moderation Effect of ED on BCT and SCI")

Reviewer #2: The writing style is very good and the topic is very interested and will attract the reader who interest in the block chain technology and the supply chain resilient. The reference list is quite update but if you can update some of the reference list to the newer one, your paper will be more interesting.

6. PLOS authors have the option to publish the peer review history of their article (what does this mean?). If published, this will include your full peer review and any attached files.

Reviewer #1: **Yes: **Asst.Prof.Dr.Wissawa Aunyawong

Reviewer #2: **Yes: **Phutthiwat Waiyawuththanapoom

---

## [Author Response · Author response to Decision Letter 0]

7 Sep 2023

Re-Submission to PLOS ONE: Does blockchain technology matter for supply chain resilience in dynamic environments? The role of supply chain integration.

Dear Reviewer 1,

Thank you very much for your review of our work and the provision of constructive comments and feedback. We hope that our modified paper has addressed your concerns and benefited from your comments and feedback to improve the quality of research reported in the paper.

Best regards

Comment 1:

- The sentence "By enabling the sharing of information, internal integration plays a crucial role [77]" (before hypothesizing H3) has too old citation (2012) since the up-to-date paper prefers the citation between 2019-2023. You may replace this citation "Aunyawong, W., Wararatchai, P., & Hotrawaisaya, C. (2020). The influence of supply chain integration on supply chain performance of auto-parts manufacturers in Thailand: a mediation approach. International Journal of Supply Chain Management, 9(3), 578-590." because this paper depict the effects of supply chain integration on resilience (flexibility), especially in automotive industry.

Reply:

Many thanks for your suggestions. Based on your feedback, the respective reference has been updated. [Location of change: Section 2.3, page 5].

Comment 2:

- In 3.1 Sampling and Data Collection, The word "logistic manager" is incorrect, "logistics manager" is correct.

Reply:

According to your suggestion, the word has been changed to become “logistics manager”. [Location of change: Section 3.1, page 7].

Comment 3:

- In 3.2 Measures, the numbers of items should be specified. (for example, 4 items of SCR were derived from...) and use "were" because each variable has several items.

Reply:

In line with this comment, the number of items used to measure each variable has been determined. [Location of change: Section 3.2, page 7]. 

Comment 4:

- 4.1 Measurement model (start with the capital letter).

Reply:

According to your suggestion, the title has been corrected, with a capital letter at the beginning of the word “Measurement”. [Location of change: Section 4.1, page 8].

Comment 5:

- The name of Table should be changed to "Direct and Moderation Effect" to suit with the info. in the table.

Reply:

Based on your suggestion, the name of Table 3 has been changed to become “Direct and Moderation Effect”. [Location of change: Section 4.2, page 10]. 

Comment 6:

- Under Table 5, correct how to write Q-Square in Line 3 (Q2 is incorrect since it can lead to misunderstand as Second Quartile)

Reply:

Based on your feedback, the abbreviation (Q-Square) has been corrected to avoid misunderstandings. [Location of change: Section 4.2, page 11]. 

Comment 7:

- The name of Figure 4 "Interactive effect of BCT and ED on SCI" is incorrect because the figure explains the moderation effect of ED on the path linking BCT and SCI. (Suggested Figure Name: "Moderation Effect of ED on BCT and SCI")

Reply:

Based on your suggestion, the name of Figure 4 has been changed to become “Moderation Effect of ED on BCT and SCI”. [Location of change: Section 4.2, page 11]. 

 

Dear Reviewer 2,

Thank you very much for your encouraging words which greatly motivate us to further improve the quality of our research paper. Moreover, thank you very much for your review of our work and the provision of your constructive feedback. Following your suggestions, we have substantially revised our manuscript to incorporate your valuable and detailed feedback.

Best regards

Comment 1:

The writing style is very good and the topic is very interested and will attract the reader who interest in the block chain technology and the supply chain resilient. The reference list is quite update but if you can update some of the reference list to the newer one, your paper will be more interesting.

Reply:

Many thanks for your suggestions. Accordingly, some of the references have been updated.

---

## [Decision Letter · Decision Letter 1]

22 Nov 2023

Does blockchain technology matter for supply chain resilience in dynamic environments? The role of supply chain integration

PONE-D-23-18950R1

Dear Dr. Al-Hakimi,

We’re pleased to inform you that your manuscript has been judged scientifically suitable for publication and will be formally accepted for publication once it meets all outstanding technical requirements.

Kind regards,

Kittisak Jermsittiparsert, Ph.D.

Academic Editor

PLOS ONE

Additional Editor Comments (optional):

Reviewers' comments:

Reviewer's Responses to Questions

**Comments to the Author**

1. If the authors have adequately addressed your comments raised in a previous round of review and you feel that this manuscript is now acceptable for publication, you may indicate that here to bypass the “Comments to the Author” section, enter your conflict of interest statement in the “Confidential to Editor” section, and submit your "Accept" recommendation.

Reviewer #1: All comments have been addressed

Reviewer #2: All comments have been addressed

2. Is the manuscript technically sound, and do the data support the conclusions?

Reviewer #1: Yes

Reviewer #2: Yes

3. Has the statistical analysis been performed appropriately and rigorously? 

Reviewer #1: Yes

Reviewer #2: Yes

4. Have the authors made all data underlying the findings in their manuscript fully available?

Reviewer #1: Yes

Reviewer #2: Yes

5. Is the manuscript presented in an intelligible fashion and written in standard English?

Reviewer #1: Yes

Reviewer #2: Yes

6. Review Comments to the Author

Reviewer #1: Thank you for revising your paper. Your article now is perfect. The topic "blockchain technology, supply chain resilience and supply chain integration" can contribute to the supply chain management studies.

Reviewer #2: The paper "Does blockchain technology matter for supply chain resilience in dynamic environments? The role of supply chain integration" is currently in the trend of the field of study roght now and the finding in this paper is very interesting. The importance of "environmental dynamism" that the authors found in the research is very crucial. It will help the lattter research and operator to enhance supply chain resilent and ensure safe supply chain operations.

The reference is also up to date and thank you for writing the decent piece of research.

7. PLOS authors have the option to publish the peer review history of their article (what does this mean?). If published, this will include your full peer review and any attached files.

Reviewer #1: **Yes: **Asst.Prof.Dr.Wissawa Aunyawong

Reviewer #2: **Yes: **Phutthiwat Waiyawuththanapoom

---

## [Editor Report · Acceptance letter]

28 Dec 2023

PONE-D-23-18950R1 

PLOS ONE

Dear Dr. Al-Hakimi, 

I'm pleased to inform you that your manuscript has been deemed suitable for publication in PLOS ONE. Congratulations! Your manuscript is now being handed over to our production team.

Kind regards, 

on behalf of

Professor Kittisak Jermsittiparsert 

%CORR_ED_EDITOR_ROLE%

PLOS ONE